# Sleep and the General Behavior of Infants and Parents during the Closure of Schools as a Result of the COVID-19 Pandemic: Comparison with 2019 Data

**DOI:** 10.3390/children8020168

**Published:** 2021-02-22

**Authors:** Yasuaki Shinomiya, Arika Yoshizaki, Emi Murata, Takashi X. Fujisawa, Masako Taniike, Ikuko Mohri

**Affiliations:** 1Molecular Research Center for Children’s Mental Development, United Graduate School of Child Development, Osaka University, Osaka 565-0871, Japan; y.shinomiya@kokoro.med.osaka-u.ac.jp (Y.S.); arika@kokoro.med.osaka-u.ac.jp (A.Y.); murata@kokoro.med.osaka-u.ac.jp (E.M.); 2Research Center for Child Mental Development, University of Fukui, Fukui 910-1193, Japan; tfuji@u-fukui.ac.jp; 3Department of Child Development, United Graduate School of Child Development, Osaka University, Osaka 565-0871, Japan; masako@kokoro.med.osaka-u.ac.jp

**Keywords:** pandemic, lifestyle, nursery school, smartphone, exercise

## Abstract

This study compared cross-sectional data from online surveys describing the sleep behavior of infants and caregivers in March 2020 (the school closure period during the early stages of the COVID-19 pandemic; n = 295, 23.8 ± 3.8 months old) and March 2019 (before the pandemic; n = 2017, 24.2 ± 3.8 months old). In comparing those two points in time, no significant differences were found in wake-up times (2019: 7:19 ± 0:46 am vs. 2020: 7:18 ± 0:47 am, *p* = 0.289), bedtimes (21:01 ± 0:48 pm vs. 21:04 ± 0:53 pm, *p* = 0.144), or nocturnal sleep times (593.7 ± 43.9 min vs. 588.1 ± 50.3 min, *p* = 0.613). Regarding the caregivers, in 2020, wake-up times (2019: 6:46 ± 0:50 am vs. 2020: 6:39 ± 0:50 am, *p* = 0.017) and bedtimes (22:53 ± 1:17 pm vs. 22:42 ± 1:04 pm, *p* = 0.016) became significantly earlier compared to 2019. Among infants staying at home, total sleep time and percentage of outdoor play decreased significantly, and media use increased significantly in 2020. Lower levels of exercise and more frequent media viewing may have caused prolonged sleep latency in these children. The percentage of caregivers responding with “negative childcare feelings” was significantly higher in the group with less than three nursery school attendance days. Caregivers and infants staying at home are a high-risk group during the pandemic.

## 1. Introduction

The spread of the novel and infectious coronavirus (hereinafter, “COVID-19”), which was first reported in Wuhan, China, in December 2019, continues to date. On 12 March 2020, the World Health Organization (WHO) declared the spread of COVID-19 a pandemic [1]. Consequently, countries worldwide began declaring states of emergency, implementing lockdowns, and establishing infection prevention controls.

On 28 January 2020, the Japanese government issued a decree declaring COVID-19 an infectious disease [2]. The government further request for nationwide school closure on 27 February, which significantly changed the overall lives of Japanese citizens. In terms of the environment surrounding young children, some nursery schools and kindergartens were closed or shortened their schooling hours, in conjunction with the government’s call for school closure [3]. The employment situation of the primary caregivers of the children also changed significantly; many engaged in remote work, experienced stagger office hours, received leaves of absence, and/or had their jobs terminated. Additionally, it became difficult to use public facilities (e.g., parks and libraries). Further, many children supposedly spent their time at home only.

The disruption of the daily way of life caused by the pandemic also affected the mental health of the children in various ways. For example, 30% of the children who experienced the 2009 swine flu (H1N1) epidemic surpassed the post-traumatic stress disorder questionnaire cutoff value [4]. Similarly, research conducted on parents with children aged between 3 and 18 years during the Shaanxi, China, COVID-19 outbreak showed an increase in symptoms corresponding to the anxiety disorder listed in the DSM-5 [5].

The pandemics and epidemics supposedly have a significant impact on sleep. During the severe acute respiratory syndrome (SARS) outbreak in 2003, symptoms such as lower sleep quality were observed among the nursing staff of designated SARS treatment hospitals [6]. Regarding the COVID-19 outbreak, delays in wake-up time and bedtime as well as an increase in sleeping hours and a decrease in sleep quality during the lockdown were observed among Italian adults. Irregular sleep-wake cycles have also been reported to correlate with depression and anxiety [7]. Other reports from Greece [8] indicate that many people reported and complained of deteriorated sleep quality from the onset of COVID-19.

Regarding the impact of the pandemic on children aged 4–10 years, a research was conducted when Italy was on lockdown. This research showed that approximately 20% of the caregivers of the subjects observed sleep issues, such as insomnia and nighttime awakening, among their children [9]. Additionally, it was reports that the wake-up- and bedtimes of children aged 2–5 years have been delayed and that the quality of their sleep deteriorated after lockdown [10]. Research conducted during the lockdown on parents with children aged 6–10 years showed that the wake-up- and bedtimes were delayed for both mothers and children. Further, the quality of sleep for both the parents and children declined; the “boredom” and “psychological difficulty of the mother” also increased [11].

Sleep is extremely important for the physical and mental development of children. Less nighttime sleeping hours for children are related to hyperactivity, lower level of cognitive function [12,13], and mental disorders [14,15]. It is also supposed that disrupted biological rhythms due to staying up late or lack of sleep can trigger autonomic dystonia and poor morning physical conditions due to hypothermia and further delay emotional development [16]. Additionally, studies have found that there is a risk that the negative impacts of poor sleep quality may persist in childhood and further extend to adulthood [17,18].

According to a survey conducted in Japan in January and late May 2020, a comparison of the lifestyle of children aged 1–2 years showed little change in wake-up time and bedtimes, while the media usage increases [19]. However, this particular study asked for the wake-up and bedtimes for only one day and did not obtain data for multiple days. It has been reported that the lifestyle of children in Japan differ between weekdays and weekends [20,21]. Consequently, our study researched the sleep habits and sleep-related lifestyle, including media usage of children for a period of eight days that included the weekend. Furthermore, this study examined the lifestyle of caregivers and any pre- and post-COVID-19 changes as caregivers’ lifestyles significantly influence the sleep of Japanese children [22].

This study compares the data related to the sleep of infants, which were collected in March 2020 (the school closure period during the early stages of the COVID-19 pandemic) and March 2019 (the period prior to the COVID-19 pandemic). Changes in the sleep state of infants and their caregivers, the living situations related to sleep, including media use and the percentage of outdoor play, and changes in psychological factors related to child care were compared. The nursery school attendance status of the children and the employment situation of the caregivers were also examined to better investigate how the impact of COVID-19 manifested in the sleep- and general lifestyles of the children.

## 2. Methods

### 2.1. Study Design

This study compared cross-sectional data from online surveys describing the sleep behavior of infants and caregivers. The data used here were originally collected to be subjected to machine learning to develop applications addressing children’s sleep habits. In such a case, the larger the dataset, the better the accuracy. This study made secondary use of these data; this resulted in a large sample size, and there was a significant difference between the sample sizes from 2019 and 2020. There were 946,000 live births in 2017 in Japan [23], and when the permissible error, reliability, and response rates were respectively set at 5%, 95%, and 50%, the required sample size for this study was estimated to be 384. 

### 2.2. Participants

An online survey was conducted after recruiting caregivers with children aged between 18–30 months through a research company (INTAGE, Inc., Tokyo, Japan), following an informed consent between the company and the participants. We only obtained data regarding sleep-associated habits and did not gather any personal information such as name, birthdate, residence, etc. This study was approved by the Graduate School of Human Sciences, Osaka University, Pedagogy Course Research Ethics Committee (Approval No.: 18058, date of approval: 21 January 2019). As part of obtaining participant consent, a research briefing was presented online to the (potential) subjects to brief them on the research purpose and general outline of the study. The subjects were deemed to have consented to the study by clicking on the provided “Agree” button.

The research periods included 3–10 March 2019, and 8–14 March 2020. In 2019 and 2020, 4912 and 612 answers were obtained, respectively.

### 2.3. Questionnaire Item

The study used the questionnaire items used in the interactive sleep awareness application “Nenne Navi^®^” [24], developed by the Osaka University. The subjects were asked to provide details pertaining the sex and age in months of their children, the age of the caregiver, the nursery school attendance status of the children, the employment status of the caregiver, the number of siblings, and their household income. Regarding sleep-related details, the respondents were asked to answer questions related to both their own and their children’s wake-up and bedtimes, time taken to fall asleep after going to bed (sleep latency), the time the children finished eating dinner and bathing, whether the children played outdoor in the day, and the start and end of the children’s nap times.

To evaluate the irregularity of individual sleep rhythms, the standard deviation (SD) as calculated from the data that had been recorded continuously for eight days. The SD was used as an index of irregularity. In addition to TV and smartphone viewing hours, the TV use termination time was asked if TV was watched from 16:00 pm onward. This measure was included as media viewing in the evening or later tends to affect sleep [25]. Regarding the smartphone use termination time, based on a previous investigation of the relationship between touchscreen use or TV exposure and sleep in 6–36-month-old infants, the impact that touchscreen use has on sleep tends to be larger than that of TV exposure [26]. Furthermore, another study showed that 12–23-month-old infants with frequent tablet and/or smartphone use suffered shorter sleep cycles and longer sleep latency [27]. Therefore, this study considers the smartphone use time per day of the children.

Regarding the feelings of the caregivers toward childcare, the respondents were asked to respond to the question “How do you feel about your child-raising today?” using a five-point scale (enjoyable, relatively enjoyable, neutral, relatively unenjoyable, and unenjoyable). Regarding the question “Did you have any of the following feelings while engaging in child-raising today?” respondents were asked to respond by making multiple selections, including “Irritated” and “Tempted to hit my child.” The respondents were required to answer these questions for eight consecutive days. The questionnaire took about 10 min to complete each day.

### 2.4. Statistical Analysis

Survey response cases where entered data were of less than four consecutive cases and/or where data omissions occurred were excluded from this study. This study only included data of at least four consecutive days, as such data were considered to be necessary and suitable to accurately ascertain the nocturnal sleep patterns of the children. There were no differences between weekdays and weekend days within the four-day clusters. Additionally, cases with underlying conditions such as atopic dermatitis and asthma, which are factors that prevent sleep [28,29], were excluded. Consequently, data from 2017 out of 4912 responses in 2019 and 295 out of 612 responses in 2020 were used for the final analysis. A t-test and/or chi-squared test was used to compare the two groups for variables such as age. A Cramér’s V was further used to measure the effect size.

For analysis, a Mann-Whitney U-test was first conducted to compare the data of the two years and examine the differences in sleep habits and lifestyles. The responses for both years were then divided and categorized into two groups regarding to the childcare arrangement -those attending nursery school for three or more days (the nursery school group) and those attending nursery school for less than three days (the staying at home group). This division was made to conduct a two-way analysis of variance (ANOVA) (i.e., 2019 or 2020, and nursery school or staying home) to further examine the differences in the sleeping habits and lifestyles of the children. The study used three days per week as a standard as in a four-day cycle, which is the minimum continuous number of input days; the possibility that one day can be categorized as a non-attendance day was considered.

A simple main effect test was conducted for variables that showed a significant difference in interaction effect. Furthermore, a two-variable correlation test was conducted for variables that showed a significant difference in interaction. The SPSS Statistics 26.0J (IBM, Armonk, NY, USA) software was used to conduct a statistical test. The significance level was set at *p* < 0.05. Regarding the feelings of the respondents toward childcare, responses such as “enjoyable,” “relatively enjoyable,” and “neutral” were regarded as “positive childcaring.” The respondents were also asked to respond to the question “Did you have any of the following feelings while engaging in child-raising today?” by making multiple selections (e.g., “Irritated” and “Tempted to hit my child”). Groups that responded with “Irritated” or “Tempted to hit my child” were categorized under “negative childcaring.”

## 3. Results

### 3.1. Participants’ Demographic Data

The ages in months of the infants studied were 24.2 ± 3.8 months old in 2019 and 23.8 ± 3.8 months old in 2020. No significant difference was found regarding this category. Additionally, no significant difference was found in the ages of the caregivers between the two years (Table 1). A chi-squared test on the sex of the children, distribution of ages in months, the childcare arrangement, the employment status of the mother, the number of siblings and annual income distribution (including tax) found no significant difference between the two years.

### 3.2. Sleep Status and General Lifestyle

#### 3.2.1. Infants Sleep Status

Table 2 presents sleep parameter data of infants for 2019 and 2020. Between the two years, no significant difference was found in terms of wake-up time (2019: 7:19 ± 0:46 am vs. 2020: 7:18 ± 0:47 am, *p* = 0.289), bedtime (21:01 ± 0:48 pm vs. 21:04 ± 0:53 pm, *p* = 0.144), nocturnal sleep time (593.7 ± 43.9 min vs. 588.1 ± 50.3 min, *p* = 0.613), the total sleep time, which includes nap and nocturnal sleep times (686.6 ± 42.7 min vs. 683.9 ± 43.4 min, *p* = 0.386), and number of times the children awakening during sleep (0.39 ± 0.67 vs. 0.35 ± 0.64, *p* = 0.086). However, sleep latency was significantly longer in 2020 (21.8 ± 14.9 min vs. 24.1 ± 18.3 min, *p* = 0.037).

#### 3.2.2. Irregularity of Infants’ Sleep

There was also no difference in sleep irregularity between the two years (Table 3).

#### 3.2.3. Leisure Activities of Infants

Regarding the leisure activity of children, the percentage of the outdoor play time of the children (64.0 ± 26.0% vs. 61.0 ± 27.1%, *p* = 0.058), TV watching time (113.6 ± 78.0 vs. 125.0 ± 87.8, *p* = 0.075), TV watching termination time (20:14 ± 1:18 pm vs. 20:08 ± 1:20 pm, *p* = 0.075), smartphone use time (14.3 ± 32.4 vs. 22.0 ± 65.6, *p* = 0.069), and smartphone use termination time (17:40 ± 2:58 pm vs. 17:19 ± 3:08 pm, *p* = 0.315) were insignificantly different between the two years. However, both the TV and smartphone use times were significantly longer in 2020 (127.9 ± 85.4 min vs.147.0 ± 112.7 min, *p* = 0.023, Table 4).

#### 3.2.4. Sleep Status of Caregivers

Regarding the sleep status of the caregivers, in 2020, their wake-up times (2019: 6:46 ± 0:50 am vs. 2020: 6:39 ± 0:50 am, *p* = 0.017) and bedtimes (22:53 ±1:17 pm vs. 22:42 ± 1:04 pm, *p* = 0.016) became significantly earlier, and their sleep latency (31.3 ± 28.8 min vs. 36.8 ± 32.0 min, *p* = 0.004) became significantly longer as compared with 2019. However, no difference was found in the nocturnal sleep time of the caregivers (442.1 ± 68.6 min vs. 440.8 ± 60.9 min, *p* = 0.647, Table 5).

#### 3.2.5. Irregularity of Caregivers’ Sleep

Regarding sleep irregularity, although no change was found in variations in sleep latency (13.0 ± 14.4 min vs. 14.4 ± 14.6 min, *p* = 0.052), variations in wake-up times (33.9 ± 21.2 min vs. 30.3 ± 17.8 min, *p* = 0.014), bedtimes (43.7 ± 29.7 min vs. 36.2 ± 22.0 min, *p* = 0.001), and nocturnal sleep (55.5 ± 28.8 min vs. 48.9 ± 23.2 min, *p* < 0.001) were all found to be smaller in 2020 (Table 6).

#### 3.2.6. The Feeling of the Caregivers

Regarding the feelings of the caregivers toward childcaring, while the proportion of positive childcare feelings increased significantly in 2020 (96.4 ± 10.9% vs. 97.4 ± 10.2%, *p* = 0.015), the proportion of negative feelings toward childcare showed no difference (26.8 ± 36.1% vs. 26.4 ± 34.3%, *p* = 0.785, Table 7).

### 3.3. Difference in the Sleep States of the Children and Caregivers Depending on the Research Year and the Childcare Arrangement

The correlation between the studied children’s sleep state variables and the childcare arrangement was examined using a two-way ANOVA. No interaction between the research year(s) and the childcare arrangement was found in relation to the wake-up times [*F* (1, 2308) = 2.26, *p* = 0.133], bedtimes [*F* (1, 2308) = 0.85, *p* = 0.357], sleep latency [*F* (1, 2308) = 2.17, *p* = 0.141], or nocturnal sleep time of the children [*F* (1, 2308) = 0.82, *p* = 0.365]. While there was an interaction between the research year(s) and the style to spend daytime for the total sleep time of the children [*F* (1, 2308) = 3.97, *p* = 0.046], the main effect of the research years [*F* (1, 2308) = 0.48, *p* = 0.488] and the style to spend daytime [*F* (1, 2308) = 3.17, *p* = 0.075] showed no significance (Table 8).

Conversely, the results of the sub-effect tests showed that the simple main effect of the research year(s) in the staying at home group was significant [*F* (1, 2308) = 3.96, *p* = 0.047] and that the total sleep time for the same group was shorter in 2020 as compared with that in 2019. However, the simple main effect of the research years was not significant for the nursery school group [*F* (1, 2308) = 1.13, *p* = 0.288]. Based on these findings, it was suggested that there was a possibility that the total sleep time of the staying at home group was impacted by COVID-19.

Further, the correlation between the sleep state variables of caregivers and the childcare arrangement was examined. The main effect for both the research year(s) and the childcare arrangement was found to be significant for wake-up times [*F* (1, 2308) = 5.80, *p* = 0.016; *F* (1, 2308) = 46.14, *p* < 0.001], bedtimes [*F* (1, 2308) = 6.41, *p* = 0.011; *F* (1, 2308) = 13.82, *p* < 0.001], and sleep latency [*F* (1, 2308) = 6.06, *p* = 0.014; *F* (1, 2308) = 8.86, *p* = 0.003]. However, no interaction was found between these variables. Furthermore, regardless the style to spend daytime, the wake-up and bedtimes of the caregivers became earlier in 2020 when compared with those in 2019. The sleep latency of the caregivers also became longer in 2020 when compared with that in 2019. Regarding the nocturnal sleep time of the caregivers, there was no significant main effect in the research year(s) and the style to spend daytime [*F* (1, 2308) = 0.00, *p* = 0.986, *F* (1, 2308) = 0.24, *p* = 0.624, respectively, Table 9].

### 3.4. Difference in the Leisure Activities of the Children Depending on the Research Year and the Childcare Arrangement

To verify whether there was a difference in the percentage of outdoor play and TV/smartphone use among the children depending on the research year and the childcare arrangement, a two-way ANOVA was conducted. The research year and the childcare arrangement were considered the independent variables. The percentage of outdoor playtime, TV watching time, TV watching termination time, smartphone use time, smartphone use termination time, and total TV/smartphone use time were used as dependent variables (Table 10). Regarding the percentage of outdoor playtime (%), the main effects of the research year and the childcare arrangement were not significant [*F* (1, 2308) = 0.48, *p* = 0.488; *F* (1, 2308) = 3.17, *p* = 0.075]; only the interaction between these variables was significant [*F* (1, 2308) = 6.85, *p* = 0.009]. A simple main effects analysis showed that the staying at home group engaged in significantly less outdoor play than the nursery school group in 2020 (*p* = 0.019), although there was no significant difference between the two variables in 2019 (*p* = 0.234). Regarding the children’s TV/smartphone use, the main effect of the childcare arrangement was significant TV viewing time and total TV/smartphone use time [*F* (1, 2308) = 55.54, *p* < 0.001; *F* (1, 2308) = 46.05, *p* < 0.001], suggesting that TV/smartphone use time was longer in the staying at home group. Next, the main effect of the year was significant for both smartphone use time and total TV/smartphone use time [*F* (1, 2308) = 11.74, *p* = 0.001; *F* (1, 2308) = 8.13, *p* = 0.004], suggesting the possibility that total smartphone use time in 2020 was longer following the impact of COVID-19. No interaction was confirmed for any of the variables (Table 10).

### 3.5. Feelings toward Childcare

To verify whether there was a difference in the feelings toward childcare depending on the research year and the childcare arrangement, a two-way analysis of variance was conducted. The research year(s) and the childcare arrangement were used as independent variables; the percentage of positive childcare feelings was used as the dependent variable. Consequently, the main effect of the childcare arrangement was significant. The staying at home group also responded with “positive childcare feelings” at a higher percentage, regardless of the year [*F* (1, 2308) = 5.27, *p* = 0.022]. Additionally, no main effect depending on the research year nor the interaction [*F* (1, 2308) = 2.89, *p* = 0.089] was found (Table 11).

As a result of examining the percentage of the caregivers that responded with “negative childcare feelings,” the study found that the main effect was similarly significant only for the childcare arrangement, where the percentage was significantly higher for the staying at home group [*F* (1, 2308) = 12.28, *p* < 0.001]. Furthermore, no main effect on the research years [*F* (1, 2308) = 0.56, *p* = 0.453] nor the interaction [*F* (1, 2308) = 0.56, *p* = 0.453] was found (Table 11).

## 4. Discussion

This study compared data on questions related to the sleep of children, which were collected in March 2020 (the early stages of the COVID-19 pandemic) and March 2019 (the period before the COVID-19 pandemic). The study conducted an analysis on the changes of the sleep status of the studied children and their caregivers, how these children spent their leisure time, and the attitude of the caregivers toward childcare. Consequently, there was no significant difference between 2019 and 2020 regarding the wake-up times, bedtimes, or nocturnal sleep patterns of the children. This finding is consistent with preceding studies [7,9]. In an international comparison, Japanese children were found to sleep less than those in other countries [30]. The National Sleep Foundation (NSF) recommends a total sleep time of 11–14 h for children aged between 1–2 years old. However, the children in this study were found to have short total sleep times in both 2019 and 2020. Furthermore, although this study indicates a significant prolongation of sleep latency, similar findings were also reported in a study that used children aged 6–10 years as subjects [9]. Generally, among small children, sleep is affected by their outdoor play time [31]. Prolonged media use has also been found to be associated with the irregular sleep rhythms of children [32,33].

Both an increasing trend in the media use of children and decreasing exercise times have been reported for those children subjected to lockdown worldwide [34]. Given that the percentage of the outdoor play decreased in 2020, although insignificantly, and that the total TV and smartphone use time of children increased significantly in the same year (Table 4), there is a possibility that such trends have impacted the sleep latency of children.

Conversely, De Georgio et al. reported that the wake-up- and bedtimes of both caregivers and children aged 2–5 years have been significantly delayed after lockdown [10]. In this study, it was found that although the wake-up- and bedtimes of the caregivers became significantly earlier in 2020, their sleep latency significantly deteriorated, while their nocturnal sleep remained unchanged (Table 5). Moreover, among caregivers, irregularities in wake-up times, bedtimes, and nocturnal sleep significantly decreased in 2020. In other words, the lives of caregivers became more regular during the COVID-19 pandemic (Table 6). Although these findings appear paradoxical, it is possible that caregivers gained more leeway regarding their time. This because their life rhythms were advanced in schedule by various factors during the COVID-19 pandemic. Unlike in Italy, which was under a stressful situation due to total lockdown, caregivers in Japan went home earlier due to the closure of schools or having to leave work for home earlier. Additionally, the working hours of Japanese caregivers also decreased [35,36]. Such time leeway is also implied in the increase in the percentage of positive childcare feelings of the caregivers in 2020 (Table 7). However, further examination is still needed.

This study also examined how the childcare arrangement impacted their sleep during COVID-19. It was found that for the children that attended a nursery school for three or more days a week, both their caregivers and the children themselves had significantly earlier wake-up times, later bedtimes, and shorter sleep latency when compared to those of children staying at home (Table 8 and Table 9). The total sleep time and the percentage of the outdoor play of the children were found to be correlated with the childcare arrangement and research years. In other words, children who staying at home suffered significantly decrease total sleep and the outdoor play in 2020 (Table 8 and Table 10). This finding was similar to the reports from Italy and Spain [34].

Notably, it has previously been reported that children who play outdoors tend to have longer total sleep times [37]. Therefore, the reported decrease outdoor play indicated in this study may have impacted the sleep of the studied children. Outdoor play time declined due to the temporary closure of 82% of all children’s recreational facility as a nationwide prevention of infectious diseases [38], a decrease in the number of places where children could play outdoors (e.g., parks) due to restrictions on the use of the respective areas, and children not being taken outdoors due to the fear of infection. However, childcare facilities were not subjected to school closer requests, even during the period during which the government ordered the closure of schools. Children in the nursery school group could, therefore, better secure opportunities for outdoor play time. This might be the reason for this study not finding a significant decrease in the percentage of outdoor play for the respective group.

Additionally, although the total TV and smartphone use times were significantly longer in 2020, children in the staying home group spent significantly longer TV viewing times in both 2019 and 2020 compared to those in the nursery school group (Table 10). According to a 2017 report, the average media contact time on weekdays was 139.0 min for one-year-old children and 167.0 min for two-year-old children; on weekends/holidays, such contact time was 179.0 min for one-year-old children and 218.0 min for two-year-old children [39]. Although this might indicate a slightly decreasing trend, further study is necessary to generate more accurate findings.

In India, a survey of people aged 18 years and older reported that an increase in the digital media use time and TV viewing time led to a decline in sleep quality [40]. Similarly, in Japan, it has been reported that approximately half of all the studied mothers use smartphones for childcare while they are busy with housework [41]. Furthermore, almost 70.4% of the mothers of all children aged 1 year have been reported to use smartphones “to soothe or calm down a sullen child” [41]. Therefore, there is a possibility that TV and smartphone use time increased during the COVID-19 pandemic due to more children becoming irritable as a result of not being able to play outdoors.

In terms of the attitude of the caregivers toward childcare, it is encouraging to note that the response of “positive childcare feelings” significantly increased in 2020. As mentioned previously, in addition to the caregivers going home earlier from work and having more free time in general, it has also previously been reported that fathers spent more time at home due to working remotely and that married couples conversed more during the pandemic [42]. Therefore, it is possible that one of the factors behind this noted increase in the positive attitude toward childcare could be the greater participation of husbands in childcare activities.

Additionally, it is ironic how the request of the government for individuals to remain indoors improved the sleep of Japanese caregivers during the pandemic. This finding may relate to the number of the Japanese caregivers who tended to work a significant amount of overtime and/or the low percentage of Japanese individuals who take paid holidays unlike in the West, where caregivers suddenly began going home earlier, resulting in improved sleep states. Conversely, many overseas countries reported an increase in domestic violence [43] and child abuse or maltreatment [44] during the COVID-19 pandemic. Although no interaction was found in this study, it should be noted that in 2020, 30% of the participating caregivers from the staying at home group had negative feelings toward childcare. Particularly, the isolation of childcare was a concern, as individuals are asked to refrain from going outside and/or were not allowed to use public facilities during the COVID-19 pandemic. Such limitations might increase the anxiety of caregivers toward childcare. Therefore, caregiver support, particularly for those who raise their children at home only, (i.e., households that do not use nursery schools), is needed.

As a result of the analysis presented in this study, a statistically significant interaction was found between the total sleeping hours and the percentage of outdoor play of the children. An especially significant change in this regard was found in families of the staying at home group. However, notably, when the Pearson’s correlation coefficient was calculated for the relationship between two variables, no significant correlation was found. This study only used the percentage of the days that the children played outdoors but did not include the number of hours children spent exercising or being outdoors. Therefore, it is necessary for future research to focus on exercise time.

In its guidelines for managing sleep disorders during lockdown, the European CBT-I Academy recommended regular sleep rhythms such as sun exposure, exercise, bedtime relaxation, media restrictions, and the provision of peace of mind, especially for mothers and their children who are considered at “high risk” of contracting the coronavirus [45]. During the early stages of the pandemic in Japan, the impact of COVID-19 on sleep rhythms was found to be insignificant. However, problems associated with lower levels of exercise and higher degrees of media viewing were found to cause prolonged sleep latency. In the future, it would be necessary for individuals to arrange their sleep adequately by taking care of themselves with regard to all aspects of life, including securing a place to exercise. Furthermore, issues such as difficulty in childcare among the staying home group were found. Therefore, this study suggests the need for relevant authorities to consider school closure measures that consider the impact that such closures may have on the sleep and development of children and households in the staying at home group.

## 5. Conclusions

During the early stages of the pandemic in Japan, although the impact of COVID-19 on sleep habits was found to be rather subtle, total sleep time and the percentage of the outdoor play decreased significantly while media use time significantly increased among infants staying at home during the daytime. In addition, the caregivers of these infants showed more stress. Our study suggests that infants and caregivers staying at home should be provided special support during the pandemic.

## 6. Limitations

The reliability of such online research is considered lower than face-to-face research.

There were differences in the number of subjects who participated in the 2019 and 2020 research programs because of differences in the durations of the data collection periods. Data collection took three months in 2019; however, in 2020, the survey was conducted within one month in response to the COVID-19 emergency. There is a possibility that this study was biased toward caregivers who are familiar with the internet. This study included caregivers with children from across the same age group in both 2019 and 2020 as subjects, the subjects for the two years were not identical.

## Figures and Tables

**Table 1 children-08-00168-t001:** Demographic Data of Subjects.

	2019 (*n* = 2017)		2020 (*n* = 295)		χ2	Cramer’s V	*p*-Value ^a^
	*n*	%	*n*	%			
gender							
male	1015	50.3	148	50.2	0.002	0.001	
female	1002	49.7	147	49.8			
Age of children							
Mean ± SD	24.2 ± 3.8		23.8 ± 3.8				0.095
18–20 mo	455	22.6	74	25.1	2.901	0.035	
21–23 mo	445	22.1	69	23.4			
24–26 mo	459	22.8	70	23.7			
27–30 mo	658	32.6	82	27.8			
Age of mothers							
Mean ± SD	34.0 ± 4.6		34. 3± 4.8				0.362
Age of father							
	36.5 ± 5.9		36.6 ± 6.0				0.808
Employment status of the mother							
working	790	39.2	110	37.3	0.389	0.013	
non-working	1179	58.5	178	60.3			
no answer	48	2.4	7	2.4			
The childcare arrangement							
Nursery school	666	33.0	91	30.9	0.551	0.015	
Staying at home	1351	67.0	204	69.2			
Number of siblings							
None	854	42.3	115	39.0	2.944	0.036	
1	868	43.0	134	45.4			
2	242	12.0	35	11.9			
3	43	2.1	10	3.4			
>4	10	0.5	1	0.3			
Family income (yen)							
<2,000,000	45	2.2	7	2.4	9.661	0.065	
<4,000,000	403	20.0	46	15.6			
<6,000,000	625	31.0	89	30.2			
<8,000,000	346	17.2	54	18.3			
<10,000,000	137	6.8	20	6.8			
10,000,000<	115	5.7	11	3.7			
unknown	346	17.2	68	23.1			

^a^, student T test.

**Table 2 children-08-00168-t002:** Infants Sleep Status in 2019 and 2020.

	2019 (*n* = 2017)			2020 (*n* = 295)			Test Statistic	*p*-Value
	Mean	Median	SD	Mean	Median	SD		
Wake-up time	7:19	7:15	0:46	7:18	7:13	0:47	308,863.0	0.289
Bedtime	21:01	21:01	0:48	21:04	21:05	0:53	313,148.5	0.144
Sleep latency (min)	21.8	20.0	14.9	24.1	20.7	18.3	275,202.0	<0.05
Nocturnal sleep time (min)	593.7	591.0	43.9	588.1	584.0	50.3	302,620.0	0.613
Total sleep time (min)	686.6	685.0	42.7	683.9	685.0	43.4	288,233.5	0.386
Number of times the children awakening during sleep	0.39	0.10	0.67	0.35	0.10	0.64	279873.0	0.086
Mann-Whitney U test.								

**Table 3 children-08-00168-t003:** Sleep Irregularity of Children in 2019 and 2020.

	2019 (*n* = 2017)			2020 (*n* = 295)			Test Statistic	*p*-Value
	Mean	Median	SD	Mean	Median	SD		
variations in wake-up time (min)	27.3	24.9	14.6	27.2	24.5	15.6	292,524.0	0.642
variations in bedtime (min)	25.6	21.9	17.6	24.4	20.6	17.3	283,318.0	0.185
variations in sleep latency (min)	12.5	9.9	9.3	13.3	10.0	12.5	303,299.5	0.587
variations in nocturnal sleep time (min)	38.1	34.0	20.2	38.7	35.1	21.8	300,441.5	0.784
variations in total sleep time (min)	47.3	43.4	22.11	45.5	41.6	21.3	283,898.5	0.204
Mann-Whitney U test.								

**Table 4 children-08-00168-t004:** The Leisure Activities of Children in 2019 and 2020.

	2019			2020			Test Statistic	*p*-Value
	Mean	Median	SD	Mean	Median	SD		
outdoor play (%)	64.0	71.0	26.0	61.0	67.0	27.1	277,481.5	0.058
TV watching time (min)	113.6	99.0	78.0	125.0	103.0	87.8	316,574.5	0.075
TV watching termination time after 16:00 pm	20:14	20:20	1:18	20:08	20:20	1:20	264,538.5	0.449
smartphone use time (min)	14.3	0.0	32.4	22.0	1.0	65.6	315,470.5	0.069
smartphone use termination time	17:40	18:20	2:58	17:19	18:00	3:08	67,156.5	0.315
TV and smartphone use times (min)	127.9	116.0	85.4	147.0	120.0	112.7	321,868.5	<0.05
Mann-Whitney test (U test).								
TV watching termination time after 16:00 pm; 2019, *n* = 1923, 2020 *n* = 283, smartphone use termination time; 2019, *n* = 950, 2020, *n* = 149, others; 2019, *n* = 2017, 2020, *n* = 295								

**Table 5 children-08-00168-t005:** Sleep Status of Caregivers in 2019 and 2020.

	2019 (*n* = 2017)			2020 (*n* = 295)			Test Statistic	*p*-Value
	Mean	Median	SD	Mean	Median	SD		
Wake-up time	6:46	6:46	0:50	6:39	6:41	0:50	271,849.5	<0.05
Bedtime	22:53	22:53	1:17	22:42	22:38	1:04	271,755.5	<0.05
Sleep latency (min)	31.3	23.1	28.8	36.8	28.3	32.0	328,110.5	<0.005
Nocturnal sleep time (min)	442.1	445.0	68.6	440.8	444.0	60.9	292,598.0	0.647
Mann-Whitney test (U test).								

**Table 6 children-08-00168-t006:** Sleep Irregularity of Caregivers in 2019 and 2020.

	2019 (*n* = 2017)			2020 (*n* = 295)			Test Statistic	*p*-Value
	Mean	Median	SD	Mean	Median	SD		
variations in wake-up time (min)	33.9	29.6	21.2	30.3	27.7	17.8	271,085.5	<0.05
variations in bedtime (min)	43.7	36.7	29.7	36.2	32.2	22.0	260,684.5	<0.01
variations in sleep latency (min)	13.0	10.0	14.4	14.4	10.0	14.6	318,246.5	0.052
variations in nocturnal sleep time (min)	55.5	50.4	28.8	48.9	43.4	23.2	259,648.0	<0.001
Mann-Whitney test (U test).								

**Table 7 children-08-00168-t007:** The feelings of the caregivers toward childcaring in 2019 and 2020.

	2019 (*n* = 2017)			2020 (*n* = 295)			Test Statistic	*p*-Value
	Mean	Median	SD	Mean	Median	SD		
positive childcare feelings (%)	96.4	100.0	10.9	97.4	100.0	10.2	313,103.0	<0.05
negative childcare feelings (%)	26.8	14.0	36.1	26.4	14.0	34.3	300,295.0	0.785
Mann-Whitney test (U test).								

**Table 8 children-08-00168-t008:** Difference in the sleep states of the children depending on the research year and the childcare arrangement (a two-way analysis of variance).

	2019				2020				Main Effect		
	Nursery School		Staying at Home		Nursery School		Staying at Home		Research Year	Nursery School/Staying at Home	Interaction
	Mean	SD	Mean	SD	Mean	SD	Mean	SD			
Wake-up time	7:01	0:34	7:28	0:48	7:06	0:34	7:24	0:51	0.05	53.55 ***	2.26
Bedtime	21:06	0:39	20:59	0:51	21:13	0:38	21:00	0:57	1.46	9.49 **	0.85
Sleep latency (min)	20.2	13.5	22.6	15.4	20.4	16.8	25.8	18.7	2.67	14.28 ***	2.17
Nocturnal sleep time (min)	571.1	34.7	604.8	43.8	568.3	38.6	596.9	52.5	3.56	121.03 ***	0.82
Total sleep time (min)	680.3	39.6	689.6	43.8	685.3	41.1	683.3	44.5	0.05	1.62	3.97 *

* *p* < 0.05, ** *p* < 0.01, *** *p* < 0.001.

**Table 9 children-08-00168-t009:** Difference in the sleep states of the caregivers depending on the research year and the childcare arrangement (a two-way analysis of variance).

	2019				2020				Main Effect		
	Nursery School		Staying at Home		Nursery School		Staying at Home		Research Year	Nursery School/Staying at Home	Interaction
	Mean	SD	Mean	SD	Mean	SD	Mean	SD			
Wake-up time	6:32	0:42	6:54	0:52	6:23	0:48	6:46	0:49	5.80 *	46.14 ***	0.02
Bedtime	22:42	1:11	22:58	1:19	22:27	0:53	22:48	1:07	6.41 *	13.82 ***	0.28
Sleep latency (min)	28.3	24.3	32.8	30.6	31.8	28.1	39.0	33.5	6.06 *	8.86 **	0.51
Nocturnal sleep time (min)	441.2	65.2	442.6	70.2	444.9	60.8	439.0	61.1	0.00	0.24	0.64

* *p* < 0.05, ** *p* < 0.01, *** *p* < 0.001.

**Table 10 children-08-00168-t010:** Difference in the leisure activities of the children depending on the research year and the childcare arrangement (a two-way analysis of variance).

	2019				2020				Main Effect		
	Nursery School		Staying at Home		Nursery School		Staying at Home		Research Year	Nursery School/Staying at Home	Interaction
	Mean	SD	Mean	SD	Mean	SD	Mean	SD			
outdoor play (%)	63.0	26.3	64.5	25.8	66.4	26.3	58.6	27.1	0.48	3.17	6.85 **
TV watching time (min)	92.3	63.3	124.1	82.3	93.1	73.2	139.2	90.2	2.32	55.54 ***	1.87
TV watching termination time after 16:00pm	20:27	1:07	20:08	1:22	20:30	1:01	19:58	1:25	0.33	21.78 ***	1.22
smartphone use time (min)	11.5	25.0	15.7	35.4	23.4	98.6	21.4	43.8	11.74 **	0.18	1.46
smartphone use termination time	18:18	2:53	17:23	2:58	18:32	2:35	16:52	3:12	0.23	19.30 ***	1.63
TV and smartphone use times (min)	103.7	68.9	139.8	90.2	116.5	125.6	160.6	103.8	8.13 **	46.05 ***	0.47

** *p* < 0.01, *** *p* < 0.001. outdoor play (%); *F*(1,2308), TV watching termination time after 16:00pm; *F*(1,2202), smartphone use termination time; *F*(1,1095), others; *F*(1,2308).

**Table 11 children-08-00168-t011:** Difference in the feelings toward childcare depending on the research year and the childcare arrangement (a two-way analysis of variance).

	2019				2020				Main Effect		
	Nursery School		Staying at Home		Nursery School		Staying at Home		Research Year	Nursery School/Staying at Home	Interaction
	Mean	SD	Mean	SD	Mean	SD	Mean	SD			
positive childcare feelings (%)	97.2	9.1	96.0	11.6	98.9	6.81	96.74	11.29	2.89	5.27 *	0.47
negative childcare feelings (%)	23.4	34.1	28.4	36.8	18.2	30.8	30.0	35.2	0.56	12.28 ***	1.95

* *p* < 0.05, *** *p* < 0.001. *F*(1,2308).

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
