# Peer review of "Sleep and the General Behavior of Infants and Parents during the Closure of Schools as a Result of the COVID-19 Pandemic: Comparison with 2019 Data"

_children, 2021, doi:10.3390/children8020168_

Round 1
Reviewer 1 Report
The subject of the article "Sleep and the General Behavior of Infants and Parents during the Closure of Schools as a Result of the COVID-19 Pandemic:
Comparison with 2019 Data" is extremely important and actual. The topics is really hot.
I have the following comments and questions for the authors. There are many awkward phrases that I do not point out here; I only point out those where the meaning cannot be interpreted:
The table 2 is extremely hard to read, maximize the writing.
The paragraph 3.3 is not clear can be rewrite in more clear format.
The conclusion need to clear and specific. My recommendation is to focus on 3 short conclusions.
Please recheck the References order.
Please double check the article by a native English reader.
Thanks again for the chance of reading the article.
Author Response
Thank you very much for comments. We were encouraged by your positive comments and happy to learn that you are interested in our study. We have carefully read your comments and found them to be useful and constructive. We now resubmit a revised manuscript for your consideration for publication in Children.
Sincerely,
Ikuko Mohri, MD, PhD
D
Our responses to each of the reviewers' comments are as follow:
Reviewer 1:
- I have the following comments and questions for the authors. There are many awkward phrases that I do not point out here; I only point out those where the meaning cannot be interpreted:
We sought English proofreading before submitting and have done so again prior to resubmitting.
- The table 2 is extremely hard to read, maximize the writing.
We apologize; due to the amount of data, we originally used font size 9. We have increased all text to font size 10 and expanded the table size.
- The paragraph 3.3 is not clear can be rewrite in more clear format.
As you suggested, we have rewritten paragraph 3.3 (3.4 in revised manuscript) more clearly as below.
3.4. Difference in the leisure activities of the children depending on the research year and the childcare arrangement
To verify whether there was a difference in the percentage of outdoor play and TV/smartphone use among the children depending on the research year and the childcare arrangement, a two-way ANOVA was conducted. The research year and the childcare arrangement were considered the independent variables. The percentage of outdoor playtime, TV viewing time, TV stopping time, smartphone use time, smartphone stopping time, and total TV/smartphone use time were used as dependent variables (Table 4-3). Regarding the percentage of outdoor playtime (%), the main effects of the research year and the childcare arrangement were not significant [F (1, 2308) = 0.48, p=.488; F (1, 2308) = 3.17, p=.075]; only the interaction between these variables was significant [F (1, 2308) = 6.85, p=.009]. A simple main effects analysis showed that the staying at home group engaged in significantly less outdoor play than the nursery school group in 2020 (p = .019), although there was no significant difference between the two variables in 2019 (p = .234). Regarding the children’s TV/smartphone use, the main effect of the childcare arrangement was significant TV viewing time and total TV/smartphone use time [F (1, 2308) = 55.54, p<.001; F (1, 2308) = 46.05, p<.001], suggesting that TV/smartphone use time was longer in the staying at home group. Next, the main effect of the year was significant for both smartphone use time and total TV/smartphone use time [F (1, 2308) = 11.74, p=.001; F (1, 2308) = 8.13, p=.004], suggesting the possibility that total smartphone use time in 2020 was longer following the impact of COVID-19. No interaction was confirmed for any of the variables (Table 4-3).
- The conclusion need to clear and specific. My recommendation is to focus on 3 short conclusions.
We shortened the conclusion to the three sentences shown below.
Conclusion
During the early stages of the pandemic in Japan, although the impact of COVID-19 on sleep habits was found to be rather subtle, total sleep time and outdoor play time decreased significantly while media use time significantly increased among infants staying at home during the daytime. In addition, the caregivers of these infants showed more stress. Our study suggests that infants and caregivers staying at home should be provided special support during the pandemic.
- Please recheck the References order.
We have rechecked the references.
- Please double check the article by a native English reader.
The revised manuscript was checked by a native English reader.

Reviewer 2 Report
The paper submitted to Children MDPI Journal approaches a hot issue regarding sleep and general behaviour of infants and parents during two data during COVID-19 pandemic. The research is well structured, adjusted to pandemic situation (on-line survey), with rich data and a solid statistical analysis. There are some issues that need to addressed and adjusted.
- page 3, line 07. Online survey: did you respect the safety of the data?
- page 2, line 16. How do you explain the difference between answers between 2019 and 2020 (4921 vs. 612)?
- page 2, line 18. Can you add some information about the duration of the questionnaire? How long does it take to be completed?
- page 4, line 50. at least for consecutive years... is there a difference between weekdays vs. weekend days?
- page 5, table 2-1. Editing the table for SD, in order to have one line.
- page 6. line 97, 104, 110. You can make it more visible and easy to read if you bold or make subparagraphs for general lifestyle, sleep status of caregivers, sleep irregularity, the feeling of the caregivers.
- page 7. Subchapter 3.1? or 3.2?
- table 4-1. editing for SD, to make it in one line
- table 4-2..the same
- page 12. good limitation chapter.
Author Response
Thank you very much for comments. We were encouraged by your positive comments and happy to learn that you are interested in our study.
We have carefully read your comments and found them to be useful and constructive. We now resubmit a revised manuscript for your consideration for publication in Children.
Sincerely,
Ikuko Mohri, MD, PhD
Our responses to each of the reviewers' comments are as follow:
Reviewer 2:
- page 3, line 07. Online survey: did you respect the safety of the data?
Thank you for your comment.
We have added this information to the Participants section, as shown below.
An online survey was conducted after recruiting caregivers with children aged between 18-30 months through a research company (INTAGE, Inc., Tokyo, Japan), following an informed consent between the company and the participants. We only obtained data regarding sleep-associated habits and did not gather any personal information such as name, birthdate, residence, etc.
- page 2, line 16. How do you explain the difference between answers between 2019 and 2020 (4921 vs. 612)?
Thank you for your question. This is an important point to clarify. It took three months for data collection via the internet survey to be completed in 2019; however, in 2020, the survey was conducted within one month in response to the COVID-19 emergency.
We have added this to the Limitations section, as shown below.
- There were differences in the number of subjects who participated in the 2019 and 2020 research programs because of differences in the durations of the data collection periods. Data collection took three months in 2019; however, in 2020, the survey was conducted within one month in response to the COVID-19 emergency.
- page 2, line 18. Can you add some information about the duration of the questionnaire? How long does it take to be completed?
The questionnaire took about 10 minutes to complete each day. We have added this information to ”2.2. QuestionnaireItem” in the Methods as below.
The questionnaire took about 10 minutes to complete each day.
Method section.
- page 4, line 50. at least for consecutive years... is there a difference between weekdays vs. weekend days?
There were no differences between weekdays and weekend days within the four-day clusters. We have added this information to the Statistical Analysis as below.
There were no differences between weekdays and weekend days within the four-day clusters.
- page 5, table 2-1. Editing the table for SD, in order to have one line.
We have edited the tables carefully.
- page 6. line 97, 104, 110. You can make it more visible and easy to read if you bold or make subparagraphs for general lifestyle, sleep status of caregivers, sleep irregularity, the feeling of the caregivers.
As you suggested, we have created subparagraphs such as Infants Sleep Status, Irregularity of Infants’ Sleep, Leisure Activities of Infants, Sleep Status of Caregivers, Irregularity of Caregivers’ Sleep, and the Feeling of the Caregivers.
- page 7. Subchapter 3.1? or 3.2?
We have corrected the numbering of this subchapter.
- table 4-1. editing for SD, to make it in one line
We have edited the table carefully.
- table 4-2..the same
We have edited the table carefully.
- page 12. good limitation chapter.
We have made Limitations a chapter.
Reviewer 3 Report
Dear authors and editor,
The manuscript titled "Sleep and the General Behavior of Infants and Parents during 3 the Closure of Schools as a Result of the COVID-19 Pandemic: Comparison with 2019 Data." This study examined the lifestyle rhythms of caregivers and any pre- and post-COVID-19 changes as home lifestyles
significantly influence the sleep of Japanese children
There are many minor and major issues I'd like the authors resolve.
Abstract
1-Add the study design to the abstract.
2- Change the keywords. Delete the words "outdoor playtime". Not found in the MeSH (Medical Subject Headings).
Introduction
3-Reference the first two paragraphs of the introduction section.
Materials and Methods
4-In this section the authors should include the study design.
5-Study size: Explain how the study size was arrived at.
Results
- adequate
Discussion
- adequate
Conclusion
- adequate
Reference
- adequate
The authors explain the limitations detected in the methodology used.
- The reliability of such online research is considered lower than face-to-face research.
- There were differences in the number of subjects who participated in the 2019 and 2020 research programs.
Author Response
Thank you very much for comments. We were encouraged by your positive comments and happy to learn that you are interested in our study. We have carefully read your comments and found them to be useful and constructive. We now resubmit a revised manuscript for your consideration for publication in Children.
Sincerely,
Ikuko Mohri, MD, PhD
Our responses to each of the reviewers' comments are as follow:
Reviewer 3
Abstract
1-Add the study design to the abstract.
We have added the study design to the abstract, as shown below.
This study compared cross-sectional data from online surveys describing the sleep behavior of infants and caregivers in March 2020 (the school closure period during the early stages of the COVID-19 pandemic; n=295, 23.8 ± 3.8 months old) and March 2019 (before the pandemic; n=2,017, 24.1 ± 3.8 months old).
2- Change the keywords. Delete the words "outdoor playtime". Not found in the MeSH (Medical Subject Headings).
We have replaced the words “outdoor playtime” with “exercise.”
Introduction
3-Reference the first two paragraphs of the introduction section.
We have checked and rearranged the references.
Materials and Methods
4-In this section the authors should include the study design.
We have added a Study Design section, as recommended.
- Methods
2.1 Study Design
This study compared cross-sectional data from online surveys describing the sleep behavior of infants and caregivers. The data used here were originally collected to be subjected to machine learning to develop applications addressing children's sleep habits. In such a case, the larger the dataset, the better the accuracy. This study made secondary use of these data; this resulted in a large sample size, and there was a significant difference between the sample sizes from 2019 and 2020. There were 941,000 live births in 2017 in Japan, and when the permissible error, reliability, and response rates were respectively set at 5%, 95%, and 50%, the required sample size for this study was estimated to be 384.
5-Study size: Explain how the study size was arrived at.
We have added an explanation of the study size in Study Design, as shown above.
The authors explain the limitations detected in the methodology used.
- The reliability of such online research is considered lower than face-to-face research.
- There were differences in the number of subjects who participated in the 2019 and 2020 research programs.
We have added the limitations that you mentioned above.
- Limitations
- The reliability of such online research is considered lower than face-to-face research.
- There were differences in the number of subjects who participated in the 2019 and 2020 research programs because of differences in the durations of the data collection periods. Data collection took three months in 2019; however, in 2020, the survey was conducted within one month in response to the COVID-19 emergency.
- There is a possibility that this study was biased toward caregivers who are familiar with the internet.
- This study included caregivers with children from across the same age group in both 2019 and 2020 as subjects, the subjects for the two years were not identical.